# When Order Meets Disorder: Modeling and Function of the Protein Interface in Fuzzy Complexes

**DOI:** 10.3390/biom11101529

**Published:** 2021-10-16

**Authors:** Sophie Sacquin-Mora, Chantal Prévost

**Affiliations:** 1CNRS, Laboratoire de Biochimie Théorique, UPR9080, Université de Paris, 13 Rue Pierre et Marie Curie, 75005 Paris, France; sacquin@ibpc.fr; 2Institut de Biologie Physico-Chimique, Fondation Edmond de Rothschild, PSL Research University, 75006 Paris, France

**Keywords:** intrinsically disordered proteins, molecular modeling, fuzzy complexes, disorder and function

## Abstract

The degree of proteins structural organization ranges from highly structured, compact folding to intrinsic disorder, where each degree of self-organization corresponds to specific functions: well-organized structural motifs in enzymes offer a proper environment for precisely positioned functional groups to participate in catalytic reactions; at the other end of the self-organization spectrum, intrinsically disordered proteins act as binding hubs via the formation of multiple, transient and often non-specific interactions. This review focusses on cases where structurally organized proteins or domains associate with highly disordered protein chains, leading to the formation of interfaces with varying degrees of fuzziness. We present a review of the computational methods developed to provide us with information on such fuzzy interfaces, and how they integrate experimental information. The discussion focusses on two specific cases, microtubules and homologous recombination nucleoprotein filaments, where a network of intrinsically disordered tails exerts regulatory function in recruiting partner macromolecules, proteins or DNA and tuning the atomic level association. Notably, we show how computational approaches such as molecular dynamics simulations can bring new knowledge to help bridging the gap between experimental analysis, that mostly concerns ensemble properties, and the behavior of individual disordered protein chains that contribute to regulation functions.

## 1. Introduction

Since the resolution of the first protein crystallographic structures some sixty years ago, and following Anfinsen’s dogma [1], the assumption that protein function requires a well-defined structure has been a cornerstone of protein science. However, at the turn of the century the biological importance of intrinsically disordered proteins (IDPs), and of intrinsically disordered regions (IDRs) in proteins, became increasingly clear [2]. What was once described as the Dark Proteome [3,4], since disordered segments remained invisible in most structural approaches, progressively turned into a central element of protein activity [5,6], taking an important part in numerous cellular processes [7,8], and thus showing that function does not always rely on a well-defined structure. Nowadays, around 10% of the 10,000 structures deposited annually in the PDB comprise long disorder regions of at least 30 residues [9], providing structural biologists with a wealth of experimental data to work on.

Accounting for disorder in protein assemblies was also a slow process [10]. One of the first scenarios considered, coupled folding to binding, would limit the conformational diversity of the interacting partners to their unbound form. It took another decade to finally apprehend the concept of fuzzy complexes, where one or both partners in the protein interaction can present some structural ambiguity [11]. In our current knowledge of disordered proteins, their binding modes span a continuum, ranging from disorder-to-order transitions, with a well-defined bound state, to disordered binding, with an also disordered bound state [12,13] (see Figure 1).

Meanwhile, our understanding of the functional importance of fuzziness in protein interactions has been steadily increasing. Protein structural heterogeneity enables interactions with multiple partners, either simultaneously or consecutively [13], and weakens the sequence constraint on specificity [14,15,16]. For example, disordered histone tails serve as hubs, regulating chromatin accessibility and playing a central part in the nucleosome stability [17].

The first part of the present review presents recent developments in computational methods designed to investigate protein interfaces that include disorder. In particular, we discuss the integration of experimental information provided by various techniques. In its second part, the review focuses on two specific cases, microtubules, and homologous recombination nucleoprotein filaments, where a network of intrinsically disordered tails exerts regulatory function in recruiting partner macromolecules, proteins, or DNA and tuning the atomic level association. Notably, we show how computational approaches such as molecular dynamics (MD) simulations can bring new knowledge to help bridging the gap between experimental analysis, that mostly concerns ensemble properties, and the behavior of individual disordered protein chains that contribute to regulation functions.

## 2. Modeling Tools for Fuzzy Complexes

### 2.1. All-Atom Force Fields

The systematic comparison of classic all atom force fields commonly used for molecular dynamics simulations for modeling disordered systems highlighted several problems [18]. The first issue met with older force fields was the overstabilization of secondary structures elements, α-helices and β-sheets, thus making the observation of the unfolded states that are characteristic of IDPs difficult [19]. This problem was addressed by optimizing the backbone torsion parameters against experimental NMR data [20,21,22]. Note that when working on the reparametrization of a force field, one must also pay some special attention to the training data that are used. For example, including data from coil fragments will help improving the reparametrization of the dihedral parameters [22,23,24]. This strategy was notably applied in the Amber ff03* and ff99SB* [25], CHARMM22* [26], OPLS-AA/M [27] and OPLS3 [28] force fields. Using a training set comprising both folded and unfolded proteins is of particular importance if one wants to model biological systems where order and disorder coexist, and this approach was used when developing the ff03CMAP force field [29].

The protein–water interaction is another central issue when developing force-fields for IDPs, as they do not have the hydrophobic core with many buried nonpolar residues that is usually found in folded proteins. As a consequence, all-atom force fields would lead to the overstabilization of the collapsed molten globule state compared to the extended state [30,31]. The refinement of the protein–water interaction can be done via an adjustment of the Lennard-Jones potential parameter [21]. This led to the development of the TIP4P-D water model, which is better suited for the extended shape of IDPs [32]. Interestingly, this permitted to solve another problem encountered with older force fields, namely the overstabilizing of protein–protein interactions, which could lead to protein aggregation [33]. Despite the success achieved through these developments, some issues remain. For example, while folded, globular proteins tend to unfold upon heating, IDPs have been shown to present some temperature-induced partial folding or the formation of secondary structures [34,35,36], and this effect still has to be accurately modeled [37].

The next step for improving the accuracy of IDP-specific force fields might lie in a better description of the electrostatic and hydrogen-bonding interactions, since these polar interactions play a key role in the IDPs structural behavior. This should be handled by polarizable force fields, and many efforts have been made in that direction over the last two decades [38,39,40,41]. Polarization has been implemented in all the traditional force fields, notably with AMOEBA [42], using fluctuating charge models [43], the Drude oscillator model [44], or induced dipoles [45]. These force fields remain computationally costly, but should greatly benefit from the increase in computational efficiency provided by GPUs.

### 2.2. Alternate Protein and Solvent Models

Coarse-grain protein models allow to push further the accessible length and time-scales of the simulations by reducing the number of degrees of freedom that have to be considered during the simulation. This can be particularly useful when investigating long timescale processes such as crowding. A classic coarse-grain approach is the use of Gō-like (or native centric) models to investigate coupled folding-binding events [46,47]. Multi-state Gō-like models have also been developed to study IDPs that can bind to different partners [48]. To describe fuzzy complexes, where no folding event is associated to binding, several alternative coarse-grain models are available, which were modified to be used for IDPs [49,50,51,52]. Among them we can mention AWSEM [53], PLUM [54], OPEP [55], UNRES [56], and SYRAH [57]. Analytical approaches derived from polymer physics somehow represent an ultimate stage of coarse-graining. These can be used to describe IDPs properties [58], but the remaining challenge is to relate these properties, and notably the phase behavior, to the IDP sequence [59,60].

Even without taking into account polarization, explicit solvent remain expensive from the computational point of view. An alternative is to use an implicit solvent model, where the solvation term will only depend on the protein coordinates. A classic implicit solvation method for folded systems is the Generalized Born model [61,62]. However, it presents the same issues as the traditional all atom force fields, namely an overstabilization of the secondary structure elements and an over collapse of the disordered states [63,64,65]. The problem can be addressed by basing the solvation term on the experimental solvation free energies of functional group and weighting it as a function of the group solvent exposure. This approach was used in the EEF1 [66] and ABSINTH [67,68] models. One should however keep in mind that using an implicit solvent model also means that one no longer has access to the detail of the solvent molecules individual behavior at the protein/water interface, and in particular to the water-mediated hydrogen bonds.

### 2.3. Algorithms

While classical molecular dynamics simulations remain a first choice tool for modeling protein assemblies, the efficient sampling of the rugged conformational landscape of IDPs is a costly process as it requires the crossing of many energy barriers. In particular, coupled folding and binding of IDPs to their partners are still out of reach due to the large number of degrees of freedom as well as the extensive conformational transitions involved in the process [69]. As a consequence, numerous enhanced sampling methods have been developed that will accelerate the exploration of a disordered system’s energy surface [70].

A first strategy is to add potential energy terms that will help overcome energy barriers along the simulation and improve the conformational space sampling. This is the case of metadynamics [71,72] and multi-canonical MD [73], which were used to investigate the coupled folding and binding process in the α-MoRE-MeV–XD complex [74] and in the RAP74-FCP1 complex [75] respectively. In umbrella sampling simulations, which were used to study the formation of the c-myb-KIX complex [76], the added potential takes the shape of a harmonic constraint. One should also consider the accelerated MD approach [77], which was used to perform microsecond-long simulations showing the partial folding of the GCN4 activation domain upon binding its coactivator [78].

Another option, that can be combined with the previous one, is to use multiple replicas from parallel trajectories which will be exchanged along time [79]. The replicas will differ by temperature in T-REMD, or by the introduction of a bias in the potential (BEMD). These approaches were used for modeling the binding of the ArkA IDP with a SH3 domain [80], and of p53 on MDM2 [81].

One can also apply constraints on a subset of degrees of liberty to accelerate the sampling. Bui and McCammon used targeted MD to investigate the conformational transitions undergone by fasciculin upon binding to acetylcholinesterase [82]. In the GNEIMO approach, the high frequency bond and angle vibrations are frozen, which enables the simulation of long timescale transitions that are inaccessible with classical MD [83].

However, the question remains whether all these enhanced sampling methods can capture realistic dynamics as well as correct ensemble properties. Comparison of the conformational dynamics obtained at different timescales (from picoseconds to tens of nanoseconds) by experimental approaches and MD simulations still presents discrepancies [84].

Conformational ensembles for IDPs and fuzzy complexes can also be generated with the *Flexible-Meccano* tool, which builds multiple copies of a polypeptide chain by random sampling of the backbone dihedral angles [85], or MoMA, a Robotics- and Artificial Intelligence-based approach initially developed to sample flexible loops but that can be used for open chains as well [86]. This approach was combined with SAXS data to investigate the structure of an intramolecular fuzzy complex in the Src family kinases [87].

### 2.4. Integrating Experimental Data

Conventional experimental methods, such as X-ray crystallography, SAXS, NMR, FRET or CryoEM, are not sufficient on their own to determine the conformational ensemble that characterizes a fuzzy complex, as they will only provide mean values and a global structural signal for the system. However, they can still bring in some precious information regarding secondary structure contents, side chain orientations and the dynamics and lifetime of local residue contacts. These can be used for the pruning of a conformational ensemble generated by an unbiased simulation [20,88]. Alternatively, experimental data will be used as a set of constraints and a starting point for the modeling of molecular assemblies involving ordered and disordered proteins [13,89]. The resulting ensembles can be found in the Protein Ensemble Database (PED, https://proteinensemble.org/) (accessed on 11 October 2021), an open access repository for the deposition of structural ensembles, including IDPs [90]. In addition, the FuzDB database (http://protdyn-database.org/) (accessed on 11 October 2021) specifically focuses on fuzzy complexes [91]. It was used to develop the FuzPred method (http://protdyn-fuzpred.org) (accessed on 11 October 2021), which predicts the binding mode of disordered proteins based on their amino acid sequences and without prior knowledge of the interaction partners [92,93]. SAXS and FRET can also provide us with information regarding the size of IDPs as measured by their radius of gyration (Rg), which can be used for the training of IDP force fields [39], or for confronting MD simulations results [94,95].

Over the past decades, integrative approaches have also proved a valuable tool for deciphering protein interactions that involve one, or more, disordered partner [96,97]. For example, NMR and all-atom MD are a classic combination to study protein assemblies, with NMR parameters being used to set up the starting structures for the simulations [98]. Solvent paramagnetic relaxation enhancement (sPRE), which uses NMR with the addition of soluble paramagnetic molecules, will provide quantitative information regarding surface accessibility at atomic resolution. This data can be used to map solvent-exposed regions in protein assemblies and allows the detection of transient interactions in fuzzy complexes [99]. Tsytlonok et al. investigated the conformational dynamics of the complex formed by the IDP p27 and Cdk2/cyclin A [100]. They combined single molecule FRET and REMD to gain further insight in a multistep binding mechanism that involves conformational selection followed by local induced folding of the p27 partner. As mentioned earlier, SAXS gives us information on the shape of biomolecular objects over a wide range of sizes, and also on their oligomerization state. The fact that this technique can handle polydisperse systems makes it particularly useful when working on IDPs and numerous ensemble modeling tools based on SAXS data have been developed [101]. The metainference approach developed in the Vendruscolo laboratory permits to simultaneously determine the structure and dynamics of macromolecular systems from cryo-electron microscopy density maps [102,103]. This was applied by Brotzakis et al. to determine the conformational ensemble and the dynamics of the tau-microtubule complex [104], based on the Cryo-EM determined structure of this macromolecular assembly [105].

Finally, one should mention the growing use of artificial intelligence and machine learning (AI/ML) approaches for characterizing conformational ensembles in disordered systems, and integrating experimental data with simulations [106]. For example, Ramanathan et al. used a ML approach to investigate the disorder to order transition in viral proteins binding on host pro-apoptotic proteins [107]. Machine learning can also be used for the refinement of force fields parameters, by adapting these to reproduce experimental SAXS scattering profiles [108].

### 2.5. Measuring and Comparing Disorder

The traditional metrics that were developed to analyze the structure of folded proteins, such as RMSD, are no longer relevant when working on IDPs, and comparing conformational ensembles of IDPs requires the development of specific tools. Lazar et al. proposed to use distance-based metrics relying on the median and the standard deviation of inter-residue distance distributions [109]. This approach is of particular interest for partially folded proteins comprising both a structured domain and IDRs, as it enables to directly identify the protein fragments that present structural similarity. The Local Compaction Plots (LCP) [110], which show the intramolecular distance between residues separated by a fixed span along the primary sequence, represent another interesting tool for analyzing MD trajectories, as they highlight disordered and folded region in the protein, while still showing its conformational diversity along time.

## 3. Functional Role of the Fuzzy Interface in the Cell

A growing body of reported observation on fuzzy interfaces depicts a continuum of association properties that range from quasi non-selective, liquid-like interactions to highly specific interactions, resulting from already mentioned folding-upon-binding mechanisms. Liquid-like association aims at ensuring proximity between the partner macromolecules and mostly involves electrostatic or polar interactions. Disordered proteins are a major component of membraneless cellular compartments, where they participate in liquid–liquid phase separation while avoiding aggregation, via the formation of dynamic, multivalent interactions [111,112]. Interestingly, high level of disorder in fuzzy interfaces are not necessarily associated to low affinity: in the complex between the human proteins histone H1 and its nuclear chaperone prothymosin-α, large opposite net charges have been shown to confer picomolar affinity to the association in spite of the absence of defined binding sites [113].

In this section, we examine intermediate situations where disordered proteins or segments present ubiquitous motifs that can transiently associate to defined binding sites on folded protein partners. We more specifically address cases where both structurally organized and disordered regions coexist in the same protein. Typical examples are proteins that present disordered C- or N-terminal tails, largely represented among DNA-binding or DNA-processing proteins. The disordered tails in these proteins generally present a net charge. Positive tails can assist the efficiency of DNA search for specific sequences by proteins such as transcription factors. The tails can non-selectively bind to DNA and promote inter-segment cross talks in a “monkey-bar”-type mechanism [114,115]. When they bear a net negative charge, the tails can compete with DNA for binding sites [116,117] or they can bind one or more protein partners. For example, the tetrameric SSB protein that binds DNA single strands, an essential contributor of DNA replication, recombination, and repair in bacteria, functions as a recruitment platform where its four negatively charged C-terminal tails can simultaneously bind one or more proteins, thus favoring the transfer of bound DNA to these partner proteins [118,119]. Competition and recruitment mechanisms are also common in self-associating proteins that present disordered tails, such as tubulin or fibrinogen [120,121]. In those cases, the strongly charged tails actively contribute to the binding of partner subunits using fly-casting types of mechanisms, but do not participate in the protein–protein interface once the assembly is formed. In the case of tubulin associating into microtubules, the tails form molecular brushes around the microtubule lattice and participate in active or passive diffusion of proteins along the microtubule protomers [122].

The delicate balance between binding and unbinding provides the disordered terminal tails affinity tuning functions: the tails have been shown to modulate binding behaviors in response to changes in salt concentration or composition. The tail properties are also very sensitive to changes in the distribution of their charges resulting from post-translational modifications, as well as associated excluded volume modifications [121]. In what follows, we will concentrate on two examples taken from our former or present studies where charged disordered tails interact with the folded core regions of the protein they belong to or with a lattice of these protein cores. We will discuss particular physical properties, frustration, and steric adaptability, that may enable the tails to appropriately respond to changes in their environment, and also how the interplay between the folded and the disordered protein regions helps participate in the binding modulation.

### 3.1. Interactions between the C-Terminal Tails of α,β-Tubulin Dimers and the Tubulin Core

Tubulin proteins exist in the cell as dimers of α- and β-tubulin, two closely related proteins whose sequences essentially differ at the level of their disordered C-terminal tails; both tails bear a net negative charge but they differ in length and amino-acid composition. α,β-tubulin dimers are the building blocks of microtubules (MT), the largest components of the cytoskeleton, that form highways for intracellular trafficking as well as separating chromosomes during meiosis. Modeling and NMR studies have shown that in tubulin dimers, both α - and β-tails can interact with the structurally organized region of the protein dimer (the core region) in spite of the core surface potential being mainly negative [95]. The tails are also known to contribute to the formation of microtubules by favoring the proper uptake of new tubulin dimers within the tubular architecture: alternative association forms of tubulin could be observed in the absence of tails [123]. It is therefore likely that the MT tubulin tails interact with free tubulin dimers during the assembly process, thus orienting the dimers toward the desired binding geometry. This association however needs to be transient, since a large fraction of the tails (notably the longer β-tails) are released during the process and become free to interact with microtubule-binding proteins (MAPs) [121]. Similar process has been observed by AFM when fibrin proteins assemble into fibrinogen [120], while the C-terminal tails of RecA proteins have been shown to be involved in their association process into filaments [124]. These observations indicate that the ability of the disordered protein tails to bind the protein core surface but also to unbind from it is key to their function.

How exactly the tails influence auto-assembly remains to be established. Theoretical simulation of the tubulin tails binding to their associated dimeric protein cores enabled to gain insights on this question [95]. Notably, while the surface spanned by the tails during atomic molecular dynamics simulations was found compatible with ensemble observations obtained by AFM (radius of gyration), the simulation enabled proposing a finer characterization of the spatial and temporal distribution of the tails, based on specifically developed metrics using the position of the tail center of mass, together with time analysis of the contacts between tails and protein cores. This analysis revealed the presence of a handful of specific tail-binding spots, or anchors, distributed on the tubulin surface and presenting reduced surface areas. The tails develop versatile interactions with these binding spots, mostly based on electrostatic complementarity [95,121]. Interestingly, we observed that negatively charged amino-acid patches distributed along the whole β-tail (see Figure 2) can individually bind separate binding spots, and that adjacent negative patches can slide within a given anchor and exchange their binding interactions. Binding different sites on the core surface does not seem to be cooperative but rather self-exclusive, one reason being that several negative patches on the tail may not be able to simultaneously access spatially separated anchors. Another factor arises from the electrostatic potential around the tubulin dimer. Indeed, we found that the electrostatic potential partitions the space available to the tails into electronegative regions, that are strongly repulsive for the most part of the tail length, and electropositive funnels that strongly attract the negative tail patches. This situation creates tension and frustration in the bound tails, part of which needs to reside in an unfavorable, repulsive region to allow contacts to form on the tubulin surface [95]. Frustration, a tradeoff between conflicting forces within their interatomic contact network environment [125], has been identified as a critical property of IDPs or IDRs binding to their protein targets [13]. Because of unsolved conflicts at such interfaces, added to the multiplicity of binding sites, the disordered regions are prone to switching to alternate binding geometries. We propose that the concept of frustration extends to long-range interactions such as the response of protein tail conformations to the potential energy created by the protein core, coupled to the physical attachment between the tail and the protein core. Long range frustration may constitute a powerful driving force to facilitate the tail unbinding from its core protein. It is also easily tunable via changes in the salt concentration or modification of the charge distribution in the tail via post-translational modifications (PTM). Indeed, recent work from Bigman and Levy showed that PTMs tune the binding ability of the tails to the MT, a function that is also partly linked to their exclusion volume properties [121]. Recent simulations of the hepatitis B virus (HBV) Core protein, that exhibits a 33-residue long, positively charged and intrinsically disordered C-terminal tail, suggests the existence of long range frustration in the binding of the negatively charged extremity of the tail to the positively charged extremity of the HBV capsid spike, with very sparse interactions between the rest of the tail and the external surface of the spikes [126].

### 3.2. Role of the RecA Protein C-Terminal Tails in Homologous Recombination

Homologous recombination permits the faithful repair of DNA double strand breaks in the genome, by recruiting intact genomic DNA (dsDNA) with sequence similar to the damaged DNA and using that DNA to restore the lost sequence continuity. To this aim, the dsDNA complementary strand is captured by a single strand (ssDNA) from the damaged DNA, in a process called strand exchange that occurs within filaments of recombinase proteins (RecA in bacteria) [127]. Alike many proteins that process DNA, *E. coli* RecA proteins present a disordered terminal tail, here a 25-amino acids, negatively charged C-terminal tail with seven acidic amino-acids. The C-terminal tail was shown to participate in the regulation of various stages of the recombination process: the filament self-assembly, the intake of the dsDNA into the filament, and the yield of strand exchange. While all those stages can take place in the absence of the tail or with partly deleted tails, the tail has been shown to mediate the response of the process to changes in pH or in magnesium concentration [124,128,129]. Specifically, full-length tails slow the RecA self-association process but promote the formation of longer and more stable filaments on ssDNA [124]. During the search, the dsDNA intake is also slowed in the presence of the tail, but this effect is reduced by adding 2mM free Mg^2+^ ions. This observation has been related to the fact that the searched dsDNA non-specifically binds to the filament gateway [130], a region that crosses the filament groove and involves basic amino-acids from the C-terminal domains. The acidic C-terminal tails may restrict the access of the dsDNA to the gateway via electrostatic repulsion or physical steric hindrance and the added magnesium ions may reduce the electrostatic repulsion and possibly induce the formation of secondary structures in the tails, which would confine the tails in a smaller volume.

How the disordered tail influences the strand exchange process is more puzzling. In the presence of the disordered tails, addition of 5 mM magnesium ions maximizes the formation of the strand exchange product; the magnesium concentration has no effect if the tail has been deleted, indicating that the tail is fully involved in the process. It has been proposed that in condition of low magnesium concentration, the tail may compete with the incorporated dsDNA for binding to the filament secondary binding site (site II) [128,129]. In that hypothesis, the tail may stimulate dsDNA binding to site II by disengaging from that site following changes in magnesium concentration [128]; alternatively, the tail may assist unbinding of non-homologous incorporated dsDNA, thus accelerating the search process [124]. However, site II is buried in the filament interior whereas the tail extremities are situated at the periphery of the filament [131,132] (Figure 3). In order for its acidic residues to reach site II in the filament interior, the tail would need to adopt a stretched conformation along the C-terminal domain toward the filament interior. Our first exploration of the tail structural dynamics by molecular dynamics simulations did not show such behavior [129]. Instead, all seven tails of a simulated filament, made of seven RecA monomers bound to a 21-nt ssDNA, remained at the exterior of the filament during the course of two 200 ns simulation with no added or 2 mM magnesium ions. The tails partly formed helical folds and partly lied on the external surface of the filament, sometimes spanning over two consecutive monomers, but they did not penetrate into the filament interior. Recently, we further explored the tail dynamics by taking into account the perturbation induced in the filament structure by the hydrolysis of ATP molecules situated at the interface between monomers. Indeed, experimental observation of the influence of the tails on the strand exchange process was performed in conditions of ATP hydrolysis, using ATP regeneration system. Our recent modeling studies indicate that the response of the filament to ATP hydrolysis may involve important modifications in the spatial partitioning of the filament groove, which may modify the tail accessibility to the filament interior. We used our published model of a 2-turn (12 monomers) filament [133] where the central RecA-RecA interface was modified for an ADP interface (the RecA-RecA binding geometries differ whether the cofactor is ATP or ADP) as a starting point for two 100-ns molecular dynamics simulations, one with no added magnesium ions and one with 5 mM magnesium concentration. Interestingly during the simulation with 5 mM magnesium, the tail associated to the central monomer with modified interface spontaneously penetrated in the filament interior and reached the secondary DNA binding site, showing that this proposed behavior is indeed topologically and energetically possible within filament architectures associated to ATP hydrolysis (Figure 3). These preliminary simulations need to be replicated and call for further investigation in order to draw any reliable conclusions on the effects of the magnesium concentration; notably, force fields adapted not only to different levels of structural disorder but that also correctly capture magnesium ion interactions need to be tested in order to confirm the reported observations. Magnesium ions can individually mediate interactions between negative charges but can also as an ensemble contribute to weaken salt-bridge interactions, therefore contributing to order-disorder transitions. Molecular dynamics is a tool of choice for disentangling individual from ensemble effects of the magnesium ions, provided that the interactions are correctly accounted for. Our present MD observations are too preliminary to conclude about the exact role of the magnesium, nevertheless they point to topological and steric factors as additional factors for the tails to exert their control function.

## 4. Conclusions

As the importance of IDPs and IDRs biological functions is now fully acknowledged, the development of specific numerical tools that permit to properly model complex assemblies where order meets disorder is reaching a point where new information can be obtained out of numerical simulation studies. This requires that experimental information, which often consists of statistical values obtained from ensemble conformations, but can also include indirect information on the IDP or IDR response to physicochemical perturbations, is integrated to the model. A large panel of new functions is now within reach, which opens the way for exciting new explorations.

## Figures and Tables

**Figure 1 biomolecules-11-01529-f001:**
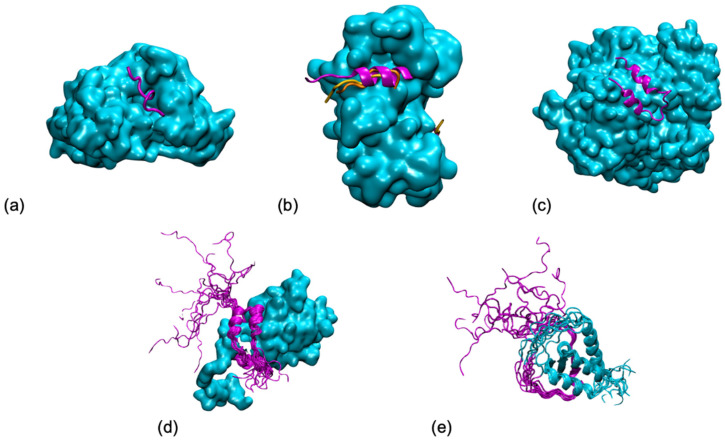
Five shades of disorder in protein/protein interactions (examples taken from (Horvath et al., PLoS Comput Biol 2020, 16, e1007864)): (**a**) Folding upon binding of an antigen (in magenta) from *P. falciparum* on an antibody (in cyan) from (pdb 4qxt). (**b**) Polymorphism of a ribosomal kinase (magenta and oranges) upon binding to S100B (in cyan) (pdb 5csf, 5csi, 5csj). (**c**) Conditional folding, the folding of the N-terminal tail (in magenta) from yeast ribonucleotide reductase (in cyan) depends on the interaction partner (pdb 1zyz). (**d**) Fuzziness, the p150 subunit of the eukaryotic initiation factor 4F (in magenta) wraps around the translation initiation factor 4E, but its N-terminal tail remains disordered (pdb 1rf8). (**e**) Disorder, both partners remain mostly disordered in the AF4-AF9 complex (pdb 2lm0). All graphical representations have been made using the VMD software.

**Figure 2 biomolecules-11-01529-f002:**
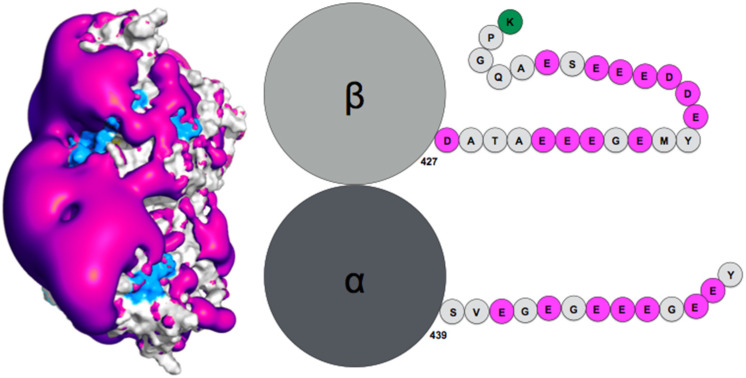
(**Left**) Negative surface electrostatic potential (-1 kT, magenta) of the αI/βIII isotype tubulin body without tails. The anchor residues involved in interactions with the disordered tail during molecular dynamics simulations are shown in blue; the representation is based on data published in (Laurin et al., Biochemistry 2017, 56, 1746); (**right**) schematic sequence of the αI/βIII isotype of tubulin, the acidic amino acids are highlighted in magenta and the basic terminal residue in green.

**Figure 3 biomolecules-11-01529-f003:**
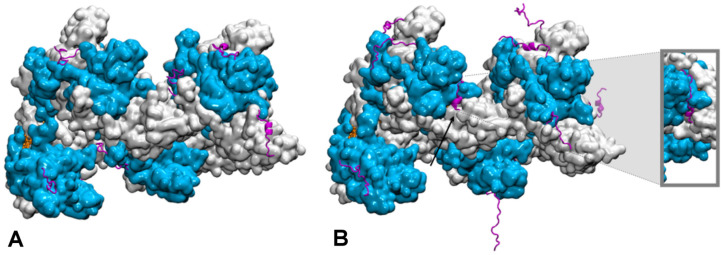
Conformational dynamics of the C-terminal tails of a two-turn, twelve monomer RecA-ssDNA filament with modified central interface, after 100 ns of molecular dynamics simulation. Successive RecA proteins are alternatively colored cyan and white. The tails (magenta, cartoon representation) explore different regions of the conformational space in terms of folding—partial α-helical folds or extended conformation—and binding to the protein core surface. (**A**) Simulation with no added salt; the tails mostly bind the core protein surface; (**B**) simulation with 5 mM Mg^2+^; some tails remain far from the surface, the tail from the central monomer (black arrow) penetrates inside the filament and reaches the filament site II, within 8 Å of the basic residue cluster of the neighboring monomer. The insert shows a view of the penetrating tail after 30° rotation around the filament axis. Simulations conditions are described in (Kim et al., Nucleic Acids Res. 2018, 46, 2548).

## Data Availability

The preliminary data presented in this study are available on request from the corresponding author.

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
