# Peer review of "When Order Meets Disorder: Modeling and Function of the Protein Interface in Fuzzy Complexes"

_biomolecules, 2021, doi:10.3390/biom11101529_

Round 1

Reviewer 1 Report

The review provides an exhaustive analysis of advancements in molecular dynamics simulations related to the need of managing fuzzy interactions. In addition the authors show a couple of biological examples in which they have applied MD to study the function of IDRs in microtubules and recombination nucleoprotein filaments. The paper is exceptionally clear in interpretation of fuzziness, relevance for biomolecular recognition problems and applications.

Minor comments:

Suggest to modify the title to (the beginning is misleading):Modeling and function of the protein interface in fuzzy complexes

Conceptually, elaborate a bit more the context-dependence of the interactions, probably the least understood aspect of fuzziness.

In paragraphs 3.1 and 3.2 make more explicit the contribution of MD and add more details to connect with section 2 (“force fields” etc.) which is very clear and exhaustive in describing what are the successful solutions to model IDRs in MD.   In particular in paragraph 3.1 (first example, microtubules) point out more what we can learn only from MD or these results complement the experimental data.

Author Response

The review provides an exhaustive analysis of advancements in molecular dynamics simulations related to the need of managing fuzzy interactions. In addition the authors show a couple of biological examples in which they have applied MD to study the function of IDRs in microtubules and recombination nucleoprotein filaments. The paper is exceptionally clear in interpretation of fuzziness, relevance for biomolecular recognition problems and applications.

We thank the reviewer for his general appreciation of our manuscript.

Minor comments

Suggest to modify the title to (the beginning is misleading): Modeling and function of the protein interface in fuzzy complexes

Even though the first part of the manuscript is more general, this review specifically focusses on the interactions between ordered and disordered protein regions and their specificities, hence the title.    

Conceptually, elaborate a bit more the context-dependence of the interactions, probably the least understood aspect of fuzziness.

The initial manuscript referred to the context-dependence of the interactions: p.6, paragr.3, "The delicate balance between binding and unbinding provides the disordered terminal tails affinity tuning functions: the tails have been shown to modulate binding behaviors in response to changes in salt concentration or composition."; or proposed possible mechanisms for this dependence  p7, last sentences "Long range frustration may constitute a powerful driving force to facilitate the tail unbinding from its core protein. It is also easily tunable via changes in the salt concentration, or modification of the charge distribution in the tail via post-translational modifications (PTM)." or p.9 "they point to topological and steric factors as additional factors for the tails to exert their control function". We have added p.9, end of last paragraph "Magnesium ions can individually mediate interactions between negative charges but can also as an ensemble contribute to weaken salt-bridge interactions, therefore contributing to order-disorder transitions."

In paragraphs 3.1 and 3.2 make more explicit the contribution of MD and add more details to connect with section 2 (“force fields” etc.) which is very clear and exhaustive in describing what are the successful solutions to model IDRs in MD. In particular in paragraph 3.1 (first example, microtubules) point out more what we can learn only from MD or these results complement the experimental data.   

We have added the following sentences to make more explicit the contribution of MD: paragr. 3.1 p.7, "Theoretical simulation of the tubulin tails binding to their associated dimeric protein cores enabled to gain insights on this question [95]. Notably, while the surface spanned by the tails during atomic molecular dynamics simulations was found compatible with ensemble observations obtained by AFM (radius of gyration), the simulation enabled proposing a finer characterization of the spatial and temporal distribution of the tails, based on specifically developed metrics using the position of the tail center of mass, together with time analysis of the contacts between tails and protein cores.  This analysis revealed the presence of a handful of specific tail-binding spots, …"; paragr. 3.2, p. 9, "notably, force fields adapted not only to different levels of structural disorder but that also correctly capture magnesium ion interactions need to be tested in order to confirm the reported observations. (…) Molecular dynamics is a tool of choice for disentangling individual from ensemble effects of the magnesium ions, provided that the interactions are correctly accounted for. Our present MD observations are too preliminary to conclude about the exact role of the magnesium, nevertheless they point to topological and steric factors as additional factors for the tails to exert their control function".

Reviewer 2 Report

I greatly enjoyed reading this lucid and comprehensive review on a very important topic in contemporary biology. The text is well written and clear. The initial introduction to "fuzziness" is useful for readers not completely familiar with the field. The review of computational methods (a key technology in this area where experimental work is often impossible) is excellent and up-to-date. The examples on tubulin and RecA provide many examples of the key concepts in action and provide an interesting overview of current thoughts about the underlying molecular mechanisms in this fascinating area of investigation.

Please find below a few grammatical corrections and some some suggestions for further improvements of an already fascinating manuscript.

-----------------------------------------------

p1, line 39:  structural biologists [plural]

p2, Figure 1 legend: I would say “(e) Disorder, 55 both partners remain mostly disordered in the AF4-AF9 complex (pdb 2lm0).” [add mostly, because there is still some folding in the interaction surface detectable]

p3, line 95: “instead of “elongated”, words like “extended” or “expanded” might be more suitable to describe such structures

p3. line 98,99: a brief description of what the temperature effect is could be useful here

p3, line 133: “water-mediated” [add hyphen?]

p3, line 135: either “simulations remain” or “simulation remains”

p4, line 144: it would also be a good idea to specifically mentioning accelerated Molecular Dynamics at this point

p4, line 190: “example” [spelling mistake]

p5, line 209 onwards: a little bit more information on AI/ML approaches would be interesting: what can these methods achieve that would be difficult to achieve with other means? Maybe also for analyzing trajectory data?

p5, line 220: the authors may also wish to have a look at a recent paper that suggests the use of “local compaction plots” based on sliding window-distance measurements to visualize various degrees of order and disorder in IDPs: Weinzierl, R. (2021). Molecular Dynamics Simulations of Human FOXO3 Reveal Intrinsically Disordered Regions Spread Spatially by Intramolecular Electrostatic Repulsion. Biomolecules 11(6):856. doi: 10.3390/biom11060856.

p.6, line: 292: are these anchors conserved in evolution, or do they vary in sequence?

Author Response

I greatly enjoyed reading this lucid and comprehensive review on a very important topic in contemporary biology. The text is well written and clear. The initial introduction to "fuzziness" is useful for readers not completely familiar with the field. The review of computational methods (a key technology in this area where experimental work is often impossible) is excellent and up-to-date. The examples on tubulin and RecA provide many examples of the key concepts in action and provide an interesting overview of current thoughts about the underlying molecular mechanisms in this fascinating area of investigation.    
Please find below a few grammatical corrections and some some suggestions for further improvements of an already fascinating manuscript.

We thank the reviewer for his comments and for his useful suggestions.

p1, line 39:  structural biologists [plural]

This has been corrected.

p2, Figure 1 legend: I would say “(e) Disorder, 55 both partners remain mostly disordered in the AF4-AF9 complex (pdb 2lm0).” [add mostly, because there is still some folding in the interaction surface detectable]

We have followed the reviewer's suggestion and have corrected the legend accordingly.

p3, line 95: “instead of “elongated”, words like “extended” or “expanded” might be more suitable to describe such structures

We have replaced "elongated" by "extended" 

p3. line 98,99: a brief description of what the temperature effect is could be useful here

We have added a short description of the temperature effects, p.3 : "For example, while folded, globular proteins tend to unfold upon heating, IDPs have been shown to present some temperature-induced partial folding or the formation of secondary structures [34-36], and this effect still has to be accurately modeled [37]"

p3, line 133: “water-mediated” [add hyphen?]

We have added the hyphen.

p3, line 135: either “simulations remain” or “simulation remains”

This has been corrected.

p4, line 144: it would also be a good idea to specifically mentioning accelerated Molecular Dynamics at this point

We thank the referee for the suggestion. Reference to accelerated dynamics have been added p.4: "One should also consider the accelerated MD approach [77], which was used to perform microsecond-long simulations showing the partial folding of the GCN4 activation domain upon binding its coactivator [78]"

p4, line 190: “example” [spelling mistake]

This has been corrected.

p5, line 209 onwards: a little bit more information on AI/ML approaches would be interesting: what can these methods achieve that would be difficult to achieve with other means? Maybe also for analyzing trajectory data?

We have added more information about the possible use of AI. The referee is right that AI can potentially become a powerful way of analyzing trajectories but so far, we have not found examples of such use in the case of disordered protein segments. 

p5, line 220: the authors may also wish to have a look at a recent paper that suggests the use of “local compaction plots” based on sliding window-distance measurements to visualize various degrees of order and disorder in IDPs: Weinzierl, R. (2021). Molecular Dynamics Simulations of Human FOXO3 Reveal Intrinsically Disordered Regions Spread Spatially by Intramolecular Electrostatic Repulsion. Biomolecules 11(6):856. doi: 10.3390/biom11060856.

We thank the reviewer for bringing this paper to our attention. We have included a reference to the LCP method, end of p 5, beginning of p. 6 : "The Local Compaction Plots (LCP) [110], which show the intramolecular distance between residues separated by a fixed span along the primary sequence, represent another interesting tool for analyzing MD trajectories, as they highlight disordered and folded region in the protein, while still showing its conformational diversity along time. "

p.6, line: 292: are these anchors conserved in evolution, or do they vary in sequence?

This is an interesting idea, however the sequence of the tubulin protein core has been mostly conserved in evolution, the great majority of sequence variability resides in the tubulin tails. Therefore, the answer is yes but this fact does not provide additional information.